# Lymphadenectomy is Unnecessary for Pure Ground-Glass Opacity Pulmonary Nodules

**DOI:** 10.3390/jcm9030672

**Published:** 2020-03-02

**Authors:** Yi-Han Lin, Chun-Ku Chen, Chih-Cheng Hsieh, Wen-Hu Hsu, Yu-Chung Wu, Jung-Jyh Hung, Po-Kuei Hsu, Han-Shui Hsu

**Affiliations:** 1Division of Thoracic Surgery, Department of Surgery, Taipei Veterans General Hospital, Taipei 100116, Taiwan; yhlin27@vghtpe.gov.tw (Y.-H.L.); cchsieh2@vghtpe.gov.tw (C.-C.H.); whhsu@vghtpe.gov.tw (W.-H.H.); wuyc@vghtpe.gov.tw (Y.-C.W.); jjhung2@vghtpe.gov.tw (J.-J.H.); pkhsu@vghtpe.gov.tw (P.-K.H.); 2Institute of Emergency and Critical Care Medicine, National Yang-Ming University, Taipei 100116, Taiwan; 3Department of Radiology, Taipei Veterans General Hospital, Taipei 100116, Taiwan; ckchen@vghtpe.gov.tw; 4Department of Surgery, New Taipei City Hospital, Sanchong, New Taipei City 24141, Taiwan

**Keywords:** ground-glass opacity, lung tumor, lymph node, lymphadenectomy, staging

## Abstract

Background: Lobectomy plus lymph node dissection is the standard treatment of early-stage lung cancer, but the low lymph node metastasis rate with ground-glass opacity (GGO) makes surgeons not perform lymphadenectomy. This study aimed to re-evaluate the lymph node metastasis rate of GGO to help make a clinical judgment. Methods: We performed this retrospective study to enroll patients who received lung cancer surgery from 2011 to 2016. Patient characteristics collected included tumor size, solid part size and lymph node metastasis rate. These patients were categorized into pure GGO and part solid GGO groups to undergo analysis. Results: Lymph node metastasis rates were 0%, 3.8% and 6.9% in order of the pure GGO group, the GGO predominant group and the solid predominant group. In the lobectomy patients, the solid predominant group still showed to have the highest lymph node metastasis rate and recurrence rate (8.3% and 10.1%). Conclusion: It is unnecessary to perform lymphadenectomy for patients with pure GGO in view of the 0% lymph node metastasis rate. The higher lymph node metastasis rate in the patients with the solid predominant group, 6.9%, suggested that surgeons should choose a rational lymphadenectomy method according to their GGO property and clinical judgment.

## 1. Introduction

Lung cancer is the leading cause of cancer all over the world and in Taiwan, and is characteristic of high mortality due to early metastasis and delayed diagnosis [1]. Air pollution, tobacco use and family history are the possible risk factors of lung cancer [1,2]. People exposed to these risk factors can now benefit from lung cancer screening with chest computed tomography (CT) [2,3]. Nodules suspected of lung cancer require surgical resection as soon as possible since only patients at the early stage may expect long-term survival with relatively low recurrent rates [2,4].

A nodule may appear on low-dose chest CT as a ground-glass opacity (GGO) having less invasive components. On the contrary, the appearance of solid components on CT images is associated with the invasive behavior of lung cancer [5]. Among the different subtypes of lung cancer, adenocarcinoma in situ (AIS), minimally invasive adenocarcinoma (MIA) and lepidic-predominant adenocarcinoma are considered to be the least invasive subtypes of adenocarcinoma with slow growth and low metastasis rate [6,7,8]. Patients are usually suggested to take regular chest CT follow-up until there is any change in tumor size or GGO components [2,8].

A low lymph node metastasis rate has been reported in GGO predominant nodules [9]. According to the National Comprehensive Cancer Network (NCCN) guideline version 2.2019, lobectomy combined with complete lymph node dissection is the golden standard treatment of early-stage lung cancer [2]. The endobronchial biopsy technique was applied to exam the lymph node metastasis condition but it is an uncomfortable procedure for patients [10]. Plenty of studies have reported that sublobar resection may have a similar oncologic outcome and recurrence rate for stage I NSCLC to preserve more pulmonary function and provide a better quality of life after surgery [11,12]. Sublobar resection seems to be an acceptable procedure for early-stage lung cancer [2,11].

Some other studies on the complete lymph node dissection of lung cancer have reported that the harvest of more lymph nodes and stations provides better recurrent-free survival and overall survival in early-stage NSCLC [13,14,15,16,17]. However, the lymph node metastasis rates in patients with the GGO component of nodules are relatively low [18]. This outcome is different from the findings that the harvest of more lymph nodes suggests better survival [13,19]. A possible reason is that the previous studies might have collected a mix of stage 1 lung cancer patients with GGO and/or solid nodules that might have a different recurrence rate and invasive component ratio [5,20]. A solid nodule may indicate a higher risk of lymph node metastasis. While it remains unclear whether complete or radical lymph node dissection is needed for stage I lung cancer with GGO components, in this study, we aimed to investigate the outcome and extent of lymph node metastasis in stage I lung cancer patients with a different GGO component of nodules.

## 2. Experimental Section

### 2.1. Data Source

We performed this retrospective cohort study to evaluate the lymph node metastasis rate in stage I lung cancer patients with GGO components. The patients selected were those who received surgery for lung cancer in Taipei Veteran General Hospital from 2011 to 2016. The study was approved by the institutional review board of Taipei Veterans General Hospital (approval number: 2020-03-008AC).

### 2.2. Radiography

The chest CT images were analyzed with a high-frequency algorithm and examined with a window level of -550 Hounsfield unit (HU) and a window width of 1600 HU as the lung window. The mediastinal window was defined as window level 40 HU and window width 400 HU. GGO nodules were divided into pure GGO and part-solid GGO groups by Suzuki’s classification [21]. Class I or II GGOs were categorized as the pure GGO group, and class III, IV or V GGOs were categorized as the part solid GGO group. Class VI, the so-called pure solid nodule, was excluded from study. The part solid GGOs were further separated into the GGO predominant group and solid predominant group by their solid part ratio. In the GGO predominant group, the solid component of a tumor had a diameter of ≤50% for the whole tumor. In the solid predominant group, the solid component accounted for 51%–100% diameter of the whole tumor.

### 2.3. Participants

In the surgery patients, metastatic carcinomas such as colon cancer or osteosarcoma with lung metastasis were excluded. In primary lung cancer patients, those who had a solid tumor or recurrent lung cancer had no pre-operative chest CT image, or expired during hospitalization, were excluded. Patients who received pre-operative chemotherapy or radiotherapy were also excluded. The surgery criteria were the GGO growing more than 20% during follow-up or emergence of the solid part. The stationary GGO would be followed up with chest CT annually. All patients eligible for lobectomy received lobectomy, and those with comorbidities received sublobar resection. 

### 2.4. Outcomes

The demographic data were collected retrospectively, including age, gender, smoking history, family history, tumor location, tumor size, solid part size, clinical N stage, operative methods, the number of lymph nodes and stations harvested, tissue histology, recurrence rate and disease-related death. The peripheral GGO was defined as the outer one-third of the lung parenchyma and the central GGO was defined as the inner two-thirds of the lung parenchyma [2]. The positron emission tomography/computed tomography (PET/CT) was arranged for excluding the incidental lymph node or distant metastasis. Patients’ tumor staging was based on the International Association for the Study of Lung cancer: 8th edition of the Classification of Malignant Tumors (TNM) classification for lung cancer [22]. For those who had clinical stage more than stage Ib or suffered from cancer recurrence during follow-up, further surgical resection, chemotherapy, target therapy or radiotherapy would be arranged depending on individual patient disease status.

### 2.5. Statistical Analysis

Categorical data were reported as counts and percentages and compared with Pearson’s chi-squared test or Fisher’s Exact Test. Continuous variables were given as mean ± standard deviation and compared using Analysis of variance (ANOVA) and POST Hoc multiple comparison with the lysergic acid diethylamide (LSD) method between groups. The Kapla–Meier method and log-rank test were applied to compare the survival curve and cancer-specific survival between these groups. Multiple Cox regression analysis was used to compare the variables including age ≤60 years or not; family history or not; GGO property including pure GGO, GGO predominant nodules and solid predominant nodules and lymph node numbers ≤15 or not to compare the differences of lymph node metastasis between these groups. *P*-value < 0.05 was considered significant. Data analysis was performed using IBM SPSS software (IBM Corp. Released 2017. IBM SPSS Statistics for Windows, Version 25.0. Armonk, NY: IBM Corp).

## 3. Results

### 3.1. Baseline Characteristics

Among a total of 2647 pulmonary carcinoma patients who received surgery during the study period, 690 were with metastatic lung cancer and 1957 patients had primary lung cancer. Based on the exclusion criteria as aforementioned, the presence of solid tumor, lack of pre-operation chest CT, recurrent lung cancer, expiration during hospitalization, administration of pre-operation chemotherapy or radiotherapy, we finally enrolled 768 GGO patients, including 325 with pure GGO, 210 with GGO predominant GGO and 233 with solid predominant GGO (Figure 1). The mean follow-up period was 43.2 months.

In these three groups, the pure GGO patients tended to be younger (57.2 ± 10.4, 62.1 ± 9.8 and 64.4 ± 10.1 years, *p* < 0.001) and have family history (21.2%, 10.0% and 7.3%, *p* < 0.001). The GGOs were majorly located at the peripheral part showing no statistical difference among them (68%, 67.1% and 68.7%, *p* = 0.943). Compared with the pure GGO group, the GGO predominant group and solid predominant group had a larger tumor and solid part (1.1 ± 0.5, 1.9 ± 0.8 and 2.2 ± 0.9 cm, *p* < 0.001; 0, 0.5 ± 0.4 and 1.6 ± 0.7 cm, *p* < 0.001). There were 84, 102 and 150 patients in the pure GGO group, GGO predominant group and solid predominant group who received PET/CT examination, and the GGO predominant group had a lower lymph node metastasis rate (5.5%, 1.9% and 5.2%, *p* = 0.344). Most pure GGO patients received sublobar resection (72.0%, 47.1% and 27.5%, *p* < 0.001). The pure GGO group harvested fewer lymph nodes than the other groups did, including the number of stations (3.4 ± 1.6, 4.3 ± 1.7 and 4.6 ± 1.5) and the number of lymph nodes (10.8 ± 7.8, 13.3 ± 7.2 and 16.0 ± 8.8). AIS, MIA and lepidic-predominant adenocarcinoma subtypes accounted for 90.5% in the pure GGO group, but only 54.3% in the GGO predominant group and 20.2% in the solid predominant group, *p* < 0.001. During the follow-up period, 26 patients suffered from recurrence and 15 patients died from related disease; most of them were in the solid predominant group (Table 1).

### 3.2. Characteristics of GGO Patients with Lobectomy Resection

To standardize the operative method of the part solid GGO group, we excluded the patients with sublobar resection. Lobectomy was performed in 91 patients in the pure GGO group, 122 patients in the GGO predominant group and 169 patients in the solid predominant group. The pure GGO group was younger than other groups (57.9 ± 9.0, 60.2 ± 9.9 and 63.5 ± 9.5, *p* < 0.001) and had a higher family history ratio (20.9%, 9.8% and 7.1%, *p* = 0.003). The GGOs were majorly located at peripheral part showing no statistical difference among these groups. The tumor and its solid part were significantly larger in the solid predominant group (2.4 ± 0.9 cm and 1.7 ± 0.7cm) than in the other two groups. There were 40, 69 and 117 patients who received pre-operative PET/CT examination, and the pure GGO group had the most lymph node metastasis in pre-operative staging (8.8%, 2.5% and 4.8%, *p* = 0.458). The ratios of invasive-predominant adenocarcinoma were 16.5%, 51.6% and 80.5%, respectively, in the pure GGO, GGO predominant and solid predominant groups, indicating significantly stronger invasiveness in the solid predominant group. Similarly, the lymph node metastasis rates of these groups, 0%, 5.0% and 8.3%, respectively, tended to increase significantly with the solid part ratio. During the follow-up period, cancer recurrence afflicted no patient (0%) in the pure GGO group, one patient (0.8%) in the GGO predominant group and 17 patients (10.1%) in the solid predominant group. The disease-related death rates of these groups were 0%, 0.8% and 5.9%. The solid predominant group obviously ran a significantly higher risk in cancer recurrence and disease-related death (Table 2).

### 3.3. Multiple Cox Regression of GGO Patients with Lobectomy Resection

A multiple Cox regression analysis was performed in the lobectomy patients to evaluate the risk factors of lymph node metastasis. Hazard ratio (HR) and confidence interval (CI) are presented in Table 3. Age >60 years and family history showed to be the two significant risk factors in the single variant analysis; however, further multiple Cox regression showed neither of them to be significant in determining the risk of lymph node metastasis (HR: 0.622, *p* = 0.287; HR: 1.003, ρ = 0.996). The GGO predominant group and solid predominant group had a higher lymph node metastasis rate (HR: 4.842, *p* = 0.146 and HR: 8.197, *p* = 0.045). The number of lymph node harvested showed an elevated hazard ratio in the lymph node > 15 group (HR: 3.564, ρ value = 0.023) (Table 3).

## 4. Discussion

The clinical significance of the GGO component is not well understood. In early-stage NSCLC, an anatomic resection may reveal a low recurrence rate and high mediastinal lymph node metastasis rate up to 25.8% [23]. Plenty of studies have demonstrated the superiority of radical lymph node dissection over lymph node sampling to guarantee a better long-term survival and lower recurrence rate [2,13,15,24,25]. However, there are still many thoracic surgeons, ranging from 28.8% to 44.6%, who do not perform lymphadenectomy for stage Ia NSCLC patients [15,26]. They might believe that the lymph node metastasis rate of early-stage lung cancer is relatively low, as reported by Haruki et al. to be only 9.1% in clinical stage I NSCLC [27]. Some other studies have evidenced no lymph node metastasis in GGO predominant early-stage adenocarcinoma [9,18]. It still remains debatable whether lymph node sampling or even dissection in GGO predominant early-stage NSCLC is necessary. 

In our study, those who had a suspicious lymph node or distant metastasis would receive PET/CT for further evaluation [28]. Under PET/CT examination, the GGO predominant group had a lower lymph node metastasis rate than other groups. This outcome was not related to actual lymph node metastasis. In the PET/CT lymph node metastasis patients, only three (9.1%) patients were diagnosed as having pathologic lymph node metastasis and the overall lymph node metastasis rate was 3.1%. The low lymph node metastasis rate was compatible with a previous report on a low lymph node metastasis rate associated with the GGO component [9,18]. The results from our study showed that the pure GGO group tended to be younger, female-predominant and exhibited a higher percentage of having a family history. Since a solid nodule is supposed to develop from a pure GGO, the lung nodule found later must have developed for quite a long period of time into components with a higher percentage of solid substance [29]. For this possible reason, female patients or those who had a family history of lung cancer may run a higher risk of developing lung cancer and are therefore in need of regular health examination to help find the lung nodules early. In the pure GGO group of patients with stage I lung cancer, our study reported no lymph node metastasis and predominantly AIS or MIA histological cell types (47.4 and 23.7%, respectively). Only 9.5% of patients of the pure GGO group were categorized as having an invasive component of predominant adenocarcinoma [29]. None of them had tumor recurrence during the follow-up period, which attributed to 0% disease-related death.

In the part solid GGO group, the lymph node metastasis rate was 3.8% in the GGO predominant group and 7.3% in the solid predominant group. The AIS and MIA ratios were 10.0% and 10.0%, respectively, in the GGO predominant group, and decreased to 0% and 4.7% in the solid predominant group. During the follow-up period, 26 patients experienced cancer recurrence (2.4% in GGO predominant group versus 9.0% in solid predominant group, *p* < 0.01). The disease-related death rates in the GGO predominant group and the solid predominant group were 1.4% versus 5.2%, *p* = 0.013 (Figure 2).

The lymph node metastasis rate in patients undergoing lobectomy was 0% in the pure GGO group, 5.0% in the GGO predominant group and 8.3% in the solid predominant group. The recurrence rates of these three groups were 0%, 0.8% and 10.1%. The pure GGO group yielded the longest cancer-specific survival of about 56.2 ± 21.7 months (Figure 3). The results showed that the solid predominant group ran a higher risk of lymph node metastasis and recurrence than the pure GGO group and GGO predominant groups did and compatible with previous studies [5].

The multiple Cox regression analysis showed that the solid part ratio > 50% was an independent risk factor of lymph node metastasis. The hazard ratio of the solid predominant group was 8.197 (*p* = 0.045). However, we found no difference in lymph node metastasis rates between pure the GGO group and the GGO predominant group (HR: 4.842, *p* = 0.146). The patients having harvested more than 15 lymph nodes showed an increased hazard ratio, compared with those having the harvest of ≤15 lymph nodes (HR: 3.564, *p* = 0.023). Maybe the lymph node positive ratio is relatively low so we should harvest more numbers for searching the metastatic lymph node [30].

Lymph node sampling or radical dissection is a controversial issue in early-stage NSCLC because of its low lymph node metastasis rate [9,14,15,17,18,19,31]. Several studies have found a 0% lymph node metastasis rate in GGO predominant nodules [9,18,27]. In our study, despite 13 patients lacking pre-operative CT and the extent of lymphadenectomy depending on surgeons’ clinical judgment, the GGO predominant group still had a 3.8% lymph node metastasis rate. A possible reason is that in previous studies pure GGO and GGO predominant nodules were categorized into the same group, thus underestimated the actual lymph node metastasis rate of GGO predominant group [9,18].

Our results demonstrated that patients with pure GGO component nodules may not need lymphadenectomy because of the 0% metastasis rate they made. Lymph node sampling may be enough for patients with GGO predominant nodules because of the low lymph node metastasis rate they revealed. Nevertheless, the somewhat higher metastasis rate rising to 7.3% in patients with solid predominant nodules that require further investigation to justify the necessity to perform lymphadenectomy in patients with GGO component small nodules.

There are some limitations to this study. First, this retrospective study was forced to exclude some patients because of profile missing. The lack of previous data may bias the outcome. Second, the extent of lymphadenectomy during operation depended on the judgment of the individualized surgeon. Similarly, the choice of radical lymph node dissection or sampling depended on the personal experience of the individualized surgeon. 

## 5. Conclusions

The present study revealed no lymph node metastasis in patients receiving lobectomy for pure GGO small nodules. In patients with GGO predominant nodules, lymph node sampling should be indicated because of the results of a mean 3.8% of lymph node metastasis rate. For patients with solid predominant part solid GGO, considering the relatively higher risk of lymph node metastasis (6.9%), radical lymph node dissection should be indicated. The necessity to perform lymphadenectomy, no matter radical dissection or sampling, requires more studies to justify the indication.

## Figures and Tables

**Figure 1 jcm-09-00672-f001:**
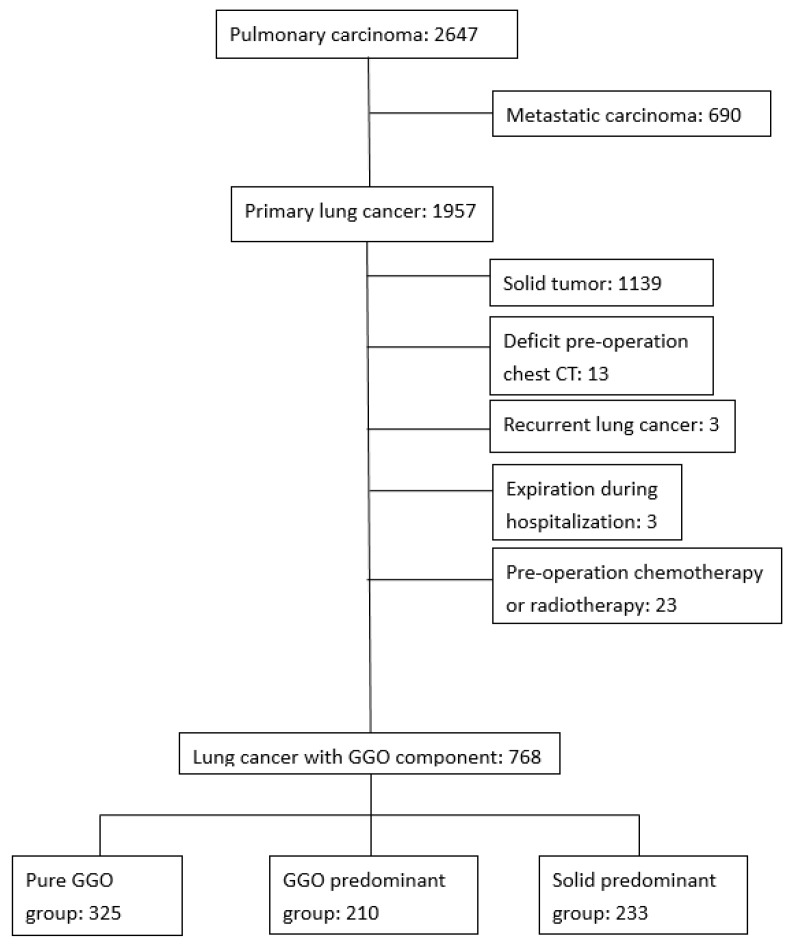
Flow diagram of the study patients. The patients were separated as pure ground-glass opacity (GGO) group, GGO predominant group and solid predominant GGO group according to their solid part diameter ratio. GGO: ground-glass opacity.

**Figure 2 jcm-09-00672-f002:**
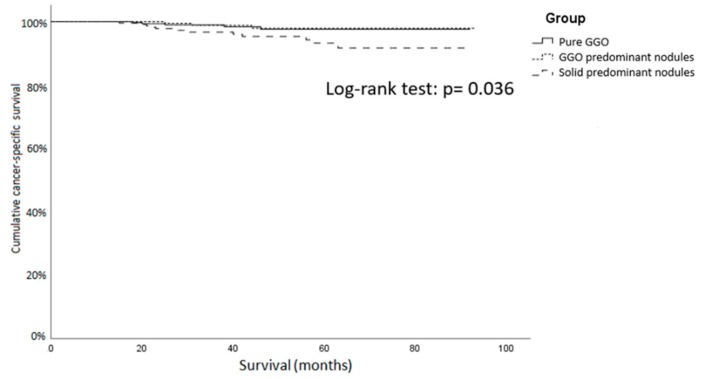
Cancer-specific survival curve of GGO patients with surgical resection. The pure GGO group and GGO predominant group showed a significantly longer cancer-specific survival than the solid predominant group. GGO: ground-glass opacity.

**Figure 3 jcm-09-00672-f003:**
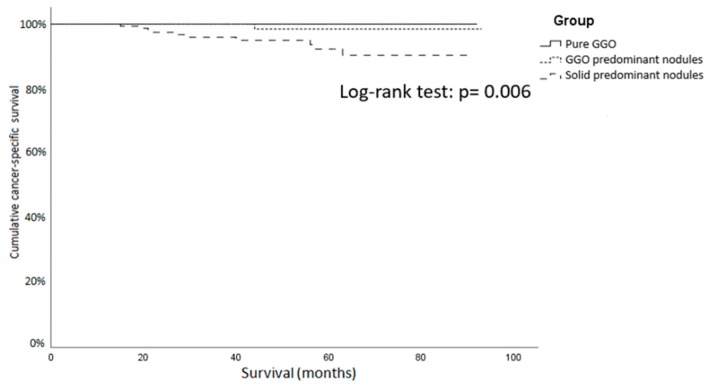
Cancer-specific survival curve of GGO patients with lobectomy or bilobectomy. The pure GGO group and GGO predominant group showed a significantly longer cancer-specific survival than the solid predominant GGO. GGO: ground-glass opacity.

**Table 1 jcm-09-00672-t001:** Demographics for patients with pure GGO or part solid GGO.

	Pure GGO (325)	GGO Predominant (210)	Solid Predominant (233)	*P*-Value
Age (Year)	57.2 ± 10.4	62.1 ± 9.8	64.4 ± 10.1	<0.001
Gender (%)				0.057
Male	103 (31.7)	90 (42.9)	102 (43.8)	
Female	212 (65.2)	120 (57.1)	131 (56.2)	
CPD (%)	126 (38.8)	90 (42.9)	109 (46.8)	0.166
Smoking (%)	52 (16.0)	33 (15.7)	50 (21.5)	0.176
Family history (%)	69 (21.2)	21 (10.0)	17 (7.3)	<0.001
Location (%)				0.943
Peripheral	221 (68%)	141 (67.1%)	160 (68.7%)	
Central	104 (32%)	69 (32.9%)	73 (31.3%)	
Tumor size (cm)				
GGO part	1.1 ± 0.5	1.9 ± 0.8	2.2 ± 0.9	<0.001
Solid part	0	0.5 ± 0.4	1.6 ± 0.7	<0.001
Clinical N stage (%)				0.344
0	307 (94.5)	206 (98.1)	222 (94.8)	
1	14 (4.3)	0	3 (1.4)	
2	4 (1.2)	4 (1.9)	8 (3.8)	
Operative method (%)				<0.001
Sublobar resection	234 (72.0)	99 (47.1)	64 (27.5)	
Lobectomy/bilobectomy	91 (28.0)	121 (52.9)	169 (72.5)	
Lymph node stations	3.4 ± 1.6	4.3 ± 1.7	4.6 ± 1.5	<0.001
N2 stations	2.3 ± 0.9	2.5 ± 0.9	2.6 ± 0.8	0.002
N1 stations	1.1 ± 1.1	1.8 ± 1.2	2.0 ± 1.1	<0.001
Lymph node number	10.8 ± 7.8	13.3 ± 7.2	16.0 ± 8.8	<0.001
N2 number	7.6 ± 6.2	8.1 ± 5.2	9.4 ± 6.4	0.003
N1 number	3.1 ± 3.9	5.2 ± 4.5	6.7 ± 4.9	<0.001
Histology (%)				<0.001
AIS	154 (47.4)	21 (10.0)	0 (0)	
MIA	77 (23.7)	21 (10.0)	11 (4.7)	
LPA	63 (19.4)	72 (34.3)	36 (15.5)	
IPA	31 (9.5)	94 (44.8)	179 (76.8)	
Other cancer	0 (0)	2 (1.0)	8 (3.4)	
T Stage (%)				<0.001
0	154 (47.4)	21 (10.0)	0 (0)	
1a	131 (40.3)	91 (43.3)	56 (24.0)	
1b	2 (0.6)	16 (7.6)	20 (8.6)	
1c	0	5 (2.4)	11 (4.7)	
2a	38 (11.7)	76 (36.2)	144 (61.8)	
4	0 (0)	1 (0.5)	2 (0.9)	
N stage (%)				0.057
0	321 (98.8)	198 (94.3)	215 (92.3)	
1	0	3 (1.4)	6 (2.6)	
2	0	5 (2.4)	10 (4.3)	
No lymphadenectomy	3 (1.2)	4 (1.9)	2 (0.9)	
Recurrence (%)	0	5 (2.4)	21 (9.0)	<0.001
Disease-related death (%)	0	3 (1.4)	12 (5.2)	0.013

GGO: ground-glass opacity; CPD: cardio-pulmonary disease; PET/CT: positron emission tomography/computed tomography; AIS: adenocarcinoma in situ; MIA: minimally invasive adenocarcinoma; LPA: lepidic-predominant adenocarcinoma; IPA: invasive-predominant adenocarcinoma.

**Table 2 jcm-09-00672-t002:** Demographics for patients receiving lobectomy or bilobectomy for small nodules with a GGO component.

	Pure GGO (91)	GGO Predominant (122)	Solid Predominant (169)	*P*-Value
Age (Year)	57.9 ± 9.0	60.2 ± 9.9	63.5 ± 9.5	<0.001
Gender (%)				0.276
Male	29 (31.9)	51 (41.8)	69 (40.8)	
Female	62 (68.1)	71 (58.2)	100 (59.2)	
CPD (%)	31 (34.1)	47 (35.8)	69 (40.8)	0.567
Smoking (%)	19 (20.9)	19 (15.6)	33 (19.5)	0.566
Family history (%)	19 (20.9)	12 (9.8)	12 (7.1)	0.003
Location (%)				0.743
Peripheral	55 (60.4)	76 (62.3)	110 (65.1)	
Central	36 (39.6)	46 (37.7)	59 (34.9)	
Tumor size (cm)				
GGO part	1.4 ± 0.6	2.1 ± 0.9	2.4 ± 0.9	<0.001
Solid part	0	0.6 ± 0.4	1.7 ± 0.7	<0.001
Clinical N stage (%)				0.458
0	83 (91.2)	119 (97.5)	161 (95.2)	
1	6 (6.6)	0	3 (1.8)	
2	2 (2.2)	3 (2.5)	5 (3.0)	
Lymph node number	17.0 ± 8.0	15.7 ± 5.8	18.3 ± 8.3	0.014
N2 number	10.0 ± 7.1	8.3 ± 4.5	10.2 ± 6.8	0.031
N1 number	6.9 ± 3.6	7.4 ± 3.8	8.1 ± 4.4	0.049
Histology (%)				<0.001
AIS	28 (30.8)	4 (3.2)	0	
MIA	20 (22.0)	10 (8.2)	3 (1.8)	
LPA	28 (30.8)	45 (36.9)	25 (14.8)	
IPA	15 (16.5)	63 (51.6)	136 (80.5)	
Other cancer	0	0	5 (3.0)	
T Stage (%)				<0.001
0	28 (30.8)	4 (3.3)	0	
1a	46 (50.5)	48 (39.3)	29 (17.2)	
1b	1 (1.1)	10 (8.2)	15 (8.9)	
1c	0	5 (4.1)	11 (6.5)	
2a	16 (17.6)	54 (44.3)	113 (66.9)	
4	0	1 (0.8)	1 (0.6)	
N stage (%)				0.041
0	91 (100)	116 (95.1)	155 (91.7)	
1	0	3 (2.5)	6 (3.6)	
2	0	3 (2.5)	8 (4.7)	
Recurrence (%)	0	1 (0.8)	17 (10.1)	<0.001
Disease-related death (%)	0	1 (0.8)	10 (5.9)	0.006

GGO: ground-glass opacity; CPD: cardio-pulmonary disease; PET/CT: positron emission tomography/computed tomography; AIS: adenocarcinoma in situ; MIA: minimally invasive adenocarcinoma; LPA: lepidic-predominant adenocarcinoma; IPA: invasive-predominant adenocarcinoma.

**Table 3 jcm-09-00672-t003:** Multiple Cox regression for the lobectomy patients with lymph node metastasis.

Variables	HR	95% CI	*p*-Value
Age			
≤60 years	-		
>60 years	0.622	0.259–1.492	0.287
Family History			
No	-		
Yes	1.003	0.231–4.366	0.996
GGO property			
Pure GGO	-		
GGO predominant	4.842	0.576–40.707	0.146
Solid predominant part solid GGO	8.197	1.052–63.862	0.045
Lymph node numbers			
≤15	-		
>15	3.564	1.193–10.648	0.023

GGO: ground-glass opacity. All the variables in the table included age, family history, GGO property and lymph node numbers.

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
