# Peer review of "Lymphadenectomy is Unnecessary for Pure Ground-Glass Opacity Pulmonary Nodules"

_jcm, 2020, doi:10.3390/jcm9030672_

Round 1

Reviewer 1 Report

In the publication “Lymphadenectomy is Unnecessary for Pure Ground Glass Opacity Pulmonary Nodules”, the authors debate about the lymphadenectomy necessity according to the appearance of the ground glass opacity in the lungs.

In my opinion, this original article is well organized and structured. The topic is interesting for the clinician community and could be helpful in the future daily practice of lung tumor resection and their surrounding nodules.

In general, the text is logical and quite easy to read despite the long names of Ground Glass Opacity (GGO) subtypes. For a better understanding, would it be possible to replace:

“GGO-predominant part-solid GGO group” by “GGO predominant group” “solid-predominant part-solid GGO group” by “solid predominant group”

As in lines 131 and 132?

I would also suggest checking the manuscript to correct some typos as the missing spaces, e.g. lines 32 or 78.

The introduction part is, in my point of view, well done and helps the reader to understand the context (as well as for pure scientists as me) of the study.

The “experimental section” part is well organized, even if I would be happy to see more information:

In the “data source” section, I would like to be ensured that the patients gave their informed consent for this work, and maybe it would be nice to have a study registration number or a study authorization. In the “participants” part, I really appreciated the description of the patients' selection In the “radiography” paragraph, for my information, would it be possible to know why the pure solid nodules (class VI) were excluded from the study? In the “outcome” section, I believe that some information about the family history would help for the text understanding. I was wondering about the nature of “the family history” until the line 164 in the conclusion part. In the “statistical analysis” part, I think that more detailed explanations would help the comprehension. Which kind of analysis was applied to which dataset? How was performed the multivariant analysis? With which variables?

I appreciated the “results” part in general and especially the tables (very helpful). I just have three remarks regarding this section:

In Figure 1: there is a misspelling in the GGO subgroups names. I can read two times “GGO-predominant part solid GGO” but the “solid-predominant part-solid GGO group” is missing. In table 3, the statistical significance is given with ρ (rho) value. But in the text, (line 206) it is reported as p (P) value. Which one is correct? I would be happy to know which statistical tests were used for the analysis in the legends of the figures/tables in order to make the results and their interpretation stronger.

The discussion and the conclusion are fine even if I still wonder if:

The distance between the tumor and the GGO was measured and if it could influence the overall survival, the disease-free survival, and the relapse events. The GGO size and location could be involved in the aggressiveness of the disease, and be linked to the metastatic events

To conclude, I appreciated this study even if I was expecting a bit more details about the statistical analyses.  

Author Response

Dear reviewer: Thank you for the opinion about our manuscript. We have revised this manuscript for a better presentation. The following is our response for your opinion.

In my opinion, this original article is well organized and structured. The topic is interesting for the clinician community and could be helpful in the future daily practice of lung tumor resection and their surrounding nodules.

Point 1: In general, the text is logical and quite easy to read despite the long names of Ground Glass Opacity (GGO) subtypes. For a better understanding, would it be possible to replace:

“GGO-predominant part-solid GGO group” by “GGO predominant group” “solid-predominant part-solid GGO group” by “solid predominant group” As in lines 131 and 132?

Response 1: The GGO-predominant part-solid GGO group was changed to GGO predominant group and the solid-predominant part-solid GGO group was changed to solid predominant group over whole manuscript.

Point 2: I would also suggest checking the manuscript to correct some typos as the missing spaces, e.g. lines 32 or 78.

Response 2: The missing spaces have been corrected. We have rechecked the whole manuscript again to ensure there is no missing spaces.

The introduction part is, in my point of view, well done and helps the reader to understand the context (as well as for pure scientists as me) of the study.

The “experimental section” part is well organized, even if I would be happy to see more information:

Point 3: In the “data source” section, I would like to be ensured that the patients gave their informed consent for this work, and maybe it would be nice to have a study registration number or a study authorization.

Response 3: The study was approved by the IRB of Taipei Veterans General Hospital. (Registration number: 2020-03-008AC) We had mentioned this in line 65~67.

Point 4: In the “participants” part, I really appreciated the description of the patients' selection In the “radiography” paragraph, for my information, would it be possible to know why the pure solid nodules (class VI) were excluded from the study?

Response 4: We believe the lymph node metastasis rate of GGOs is different from the solid nodules (which was reported by previous studies). So we want to evaluate the morphology of GGO patients in this study.

Point 5: In the “outcome” section, I believe that some information about the family history would help for the text understanding. I was wondering about the nature of “the family history” until the line 164 in the conclusion part.

Response 5: We had added the results and discussion of family history in line 124, 148, 171 and 200. We learned from the results that the pure GGO group had more predominant family history ratio, which may be because that these patients with family history may have CT examination earlier, which leads to a higher pure GGO ratio in patients with family history.

Point 6: In the “statistical analysis” part, I think that more detailed explanations would help the comprehension. Which kind of analysis was applied to which dataset? How was performed the multivariant analysis? With which variables?

Response 6: We have asked a statistician and improved the statistical method in the line100-109. Categorical data were reported as counts and percentages and compared with Pearson’s chi-squared test or Fisher’s Exact Test. Continuous variables were given as mean ± standard deviation and compared using ANOVA and POST HOC multiple comparison with LSD method between groups. Kaplan-Meier method and log-rank test were applied to compare the survival curve and cancer-specific survival between these groups. The multiple Cox-regression analysis to compare the variables including age ≦60 years or not; family history or not; GGO property including pure GGO, GGO predominant nodules and solid predominant nodules and lymph node numbers ≦15 or not to compare the differences of lymph node metastasis between these groups. P value < 0.05 was considered significant. Data analysis was performed using IBM SPSS software (IBM Corp. Released 2017. IBM SPSS Statistics for Windows, Version 25.0. Armonk, NY: IBM Corp).

Point 7: I appreciated the “results” part in general and especially the tables (very helpful). I just have three remarks regarding this section:

In Figure 1: there is a misspelling in the GGO subgroups names. I can read two times “GGO-predominant part solid GGO” but the “solid-predominant part-solid GGO group” is missing.

Response 7: We had corrected and rechecked whole manuscript again.

Point 8: In table 3, the statistical significance is given with ρ (rho) value. But in the text, (line 206) it is reported as p (P) value. Which one is correct?

Response 8: We had corrected it as the p value. The ρ (rho) was a type error.

Point 9: I would be happy to know which statistical tests were used for the analysis in the legends of the figures/tables in order to make the results and their interpretation stronger.

Response 9: We had added the statistical detail in the line 100~109. Categorical data were reported as counts and percentages and compared with Pearson’s chi-squared test or Fisher’s Exact Test. Continuous variables were given as mean ± standard deviation and compared using ANOVA and POST HOC multiple comparison with LSD method between groups. Kaplan-Meier method and log-rank test were applied to compare the survival curve and cancer-specific survival between these groups. The multiple Cox-regression analysis to compare the variables including age ≦60 years or not; family history or not; GGO property including pure GGO, GGO predominant nodules and solid predominant nodules and lymph node numbers ≦15 or not to compare the differences of lymph node metastasis between these groups. P value < 0.05 was considered significant. Data analysis was performed using IBM SPSS software (IBM Corp. Released 2017. IBM SPSS Statistics for Windows, Version 25.0. Armonk, NY: IBM Corp).

Point 10: The discussion and the conclusion are fine even if I still wonder if:

The distance between the tumor and the GGO was measured and if it could influence the overall survival, the disease-free survival, and the relapse events. The GGO size and location could be involved in the aggressiveness of the disease, and be linked to the metastatic events

Response 10: We had added the detail as tumor location (peripheral or central) to differentiate its’ relationship with lymph node metastasis at the table1, table 2, line 125 and line 148-149. The lymph node metastasis rate had no difference between the central and peripheral GGO groups.

To conclude, I appreciated this study even if I was expecting a bit more details about the statistical analyses.

Thank you for the opinion that we can improve our study for a better presentation.

Sincerely,

Yi-Han Lin, MD

Reviewer 2 Report

Thank you for submitting this article to the JCM. I was pleased to receive it as a reviewer.
I have the following questions for you, which I believe, need to be addressed before publication:
First, please justify the absence of an IRB approval.
Secondly, the paper should be written according to the STROBE (Strengthening the reporting of observational studies in epidemiology) [www.strobe-statement.org]. A STROBE checklist should also be
added [https://www.strobe-
statement.org/fileadmin/Strobe/uploads/checklists/STROBE_checklist_v4_combined.doc].
The statistical analysis should be written according to the recently published guidelines (Hickey GL, Dunning J, Seifert B, Sodeck G, Carr MJ, Beyersdorf F on behalf of the EJCTS and ICVTS Editorial Committees Editor'sChoice: Statistical and data reporting guidelines for the European Journal of Cardio-Thoracic Surgery and the Interactive CardioVascular and Thoracic Surgery. Eur J Cardiothorac Surg 2015;48:180-93).
The limitations section should be improved with a better discussion.

Besides, the discussion should be improved with a better search of the literature.
About minor points, there are grammars and typos errors in the text. Please thoroughly check the article.
Good luck with your article, and thanks again for submitting it.

Author Response

Dear reviewer: Thank you for the opinion about our manuscript. We have revised this manuscript for a better presentation. The following is our response for your opinion. Point 1: I have the following questions for you, which I believe, need to be addressed before publication:
First, please justify the absence of an IRB approval.

Response 1: The study was approved by the IRB of Taipei Veterans General Hospital. (Registration number: 2020-03-008AC) We had mentioned this in line 65~67. Point 2: Secondly, the paper should be written according to the STROBE (Strengthening the reporting of observational studies in epidemiology) [www.strobe-statement.org]. A STROBE checklist should also be
added [https://www.strobe-
statement.org/fileadmin/Strobe/uploads/checklists/STROBE_checklist_v4_combined.doc].

Response 2: We had complete the STROBE list and marked all the category position. The STROBE checking list is added in the end of the attached profile. Point 3: The statistical analysis should be written according to the recently published guidelines (Hickey GL, Dunning J, Seifert B, Sodeck G, Carr MJ, Beyersdorf F on behalf of the EJCTS and ICVTS Editorial Committees Editor'sChoice: Statistical and data reporting guidelines for the European Journal of Cardio-Thoracic Surgery and the Interactive CardioVascular and Thoracic Surgery. Eur J Cardiothorac Surg 2015;48:180-93).

Response 3: We have asked a statistician and improved the statistical method in the line100-109. Categorical data were reported as counts and percentages and compared with Pearson’s chi-squared test or Fisher’s Exact Test. Continuous variables were given as mean ± standard deviation and compared using ANOVA and POST HOC multiple comparison with LSD method between groups. Kaplan-Meier method and log-rank test were applied to compare the survival curve and cancer-specific survival between these groups. The multiple Cox-regression analysis to compare the variables including age ≦60 years or not; family history or not; GGO property including pure GGO, GGO predominant nodules and solid predominant nodules and lymph node numbers ≦15 or not to compare the differences of lymph node metastasis between these groups. P value < 0.05 was considered significant. Data analysis was performed using IBM SPSS software (IBM Corp. Released 2017. IBM SPSS Statistics for Windows, Version 25.0. Armonk, NY: IBM Corp). Point 4: The limitations section should be improved with a better discussion.

Response 4: The limitations had added in line 250-254. There was 13 patients loss of pre-operation chest CT and this may influence our result. Meanwhile, the extent of lymphadenectomy was depended on surgeon’s decision. In the table 2, we excluded the sublobar resection for making sure that patients’ received radical lymph node dissection and the N1 nodes (Station 10~13) were harvested. The lobectomy patients, as the table 2, was selected to even the selection bias that the extent of lymphadenectomy was depended on surgeons’ clinical decision.

Point 5: Besides, the discussion should be improved with a better search of the literature.

Response 5: We have added the following reference for corroborating the study result.

  1. The accuracy of integrated PET-CT compared with dedicated PET alone for the staging of patients with nonsmall cell lung cancer. In line 193 for corroborating the PET/CT is using for differenting lymph node or distant metastasis. Reference list: 29
  2. Growth of pure ground-glass lung nodule detected at computed tomography. J Thorac Dis 2015, 7. In line 202 and 207 for corroborating the GGO would grow to a solid nodule. Reference list: 30.
  3. Incidental, subsolid pulmonary nodules at CT: etiology and management. Cancer Imaging 2013, 13. In line225 for corroborating the solid component represents as the invasive component. Reference list: 5.
  4. Number of lymph nodes and metastatic lymph node ratio are associated with survival in lung cancer. Ann Thorac Surg 2012, 93. In line 236 for corroborating the low lymph node positive ratio made us needed to harvest more lymph node number. Reference: 31.

Point 6: About minor points, there are grammars and typos errors in the text. Please thoroughly check the article.

Response 6: We had corrected it and rechecked whole manuscript again. Good luck with your article, and thanks again for submitting it

Thank you for the opinion that we can improve our study for a better presentation.

Sincerely,

Yi-Han Lin, MD

Reviewer 3 Report

Thank you for submitting this article: I was pleased to receive it as a reviewer. This manuscript is well written and has some strenghts like its large size and the way in which the two groups were nicely balanced. The message is strong.

However, I have some queries that you should address first:

Major Comments

1) Maybe authors should add results of the CT scan about lymph node in order to give a cTNM classification

2) In the same way, the realization of a PET CT scan is not mentioned? it could be interesting in order to see lymph node could be detected?

3) As authors said in the manuscript , many surgeon perform sublobar resection for GGOs, a paragraph dedicated to this subgroup with the outcome could be interesting.

4) For patient who present recurence and / or die, and for patients who are classified more than Ib,  authors should mentioned if they received adjuvant treatment ( Chemotherapy or radiation therapy)

Minor comments

5)page 3 line 103 : it could be more clear to change " 223 with part solid GGOs" by "223 with solid predominant part-solid GGOs"

6) Figure 1 : the 2 box of the part solid group are the same : please change the second one for "solid predominant part-solid GGOs:223"

Author Response

Dear reviewer: Thank you for the opinion about our manuscript. We have revised this manuscript for a better presentation. The following is our response for your opinion.

Major Comments

Point 1: Maybe authors should add results of the CT scan about lymph node in order to give a cTNM classification

Response 1: In our study, all patients were categorized as clinical T1 under the chest CT scan. There were 12 patients being categorized as N1 or N2 and received further PET/CT examination. In the 12 patients, only 6 patients being diagnosed as lymph node metastasis and only 2 patients being diagnosed as pN+ later. Most patients showed incidental lymph node metastasis and there was no correlation between clinical N stage and pathologic N stage. The clinical N+ patient number was too small and no statistical difference so we did not present it in the manuscript.

Point 2: In the same way, the realization of a PET CT scan is not mentioned? it could be interesting in order to see lymph node could be detected?

Response 2: The PET/CT detail was presented in Table 1, table 2, line 128-130, 151-153. The lymph node metastasis rate over these groups were 5.5%, 1.9% and 5.2%, p = 0.344 (All patients); 8.8%, 2.5% and 4.8%, p = 0.458 (Lobectomy patients). There was no patient been categorized as M1.

Point 3: As authors said in the manuscript , many surgeon perform sublobar resection for GGOs, a paragraph dedicated to this subgroup with the outcome could be interesting.

Response 3: We believe the sublobar resection was hard to approach the station 10-14 lymph node. This influence would make us underestimate the lymph node metastasis rate of GGO patients. So we just analyze the lobectomy patients for a more precise outcome.

Point 4: For patient who present recurence and / or die, and for patients who are classified more than Ib, authors should mentioned if they received adjuvant treatment ( Chemotherapy or radiation therapy)

Response 4: The patients who are classified as more than stage Ib after surgery, would be submitted to receive further adjuvant chemotherapy, radiotherapy or target therapy depended on individual patient’s disease status.(line 95~98).

Minor comments

Point 5: page 3 line 103 : it could be more clear to change " 223 with part solid GGOs" by "223 with solid predominant part-solid GGOs"

Point 6: Figure 1 : the 2 box of the part solid group are the same : please change the second one for "solid predominant part-solid GGOs:223"

Response 5&6: We had corrected these mistakes and rechecked whole manuscript again to prevent this kind of mistake.

Thank you for the opinion that we can improve our study for a better presentation.

Sincerely,

Yi-Han Lin, MD

Round 2

Reviewer 1 Report

Dear Authors,

Thanks a lot for answering all my questions, remarks and requests.

I do not have further comments on this manuscript.

Good luck,

Reviewer 2 Report

Thank you for your answers to my comments.